# Age-Related Improvements in Peak Cardiorespiratory Fitness among Coronary Heart Disease Patients Following Cardiac Rehabilitation

**DOI:** 10.3390/jcm8030310

**Published:** 2019-03-05

**Authors:** Laura Banks, Joseph Cacoilo, Jasmine Carter, Paul I. Oh

**Affiliations:** Cardiac Prevention and Rehabilitation Program, Toronto Rehabilitation Institute, University Health Network, Toronto, ON M4G 1R7, Canada; laura.banks@uhn.ca (L.B.); josephcacoilo@gmail.com (J.C.); jasmine.carter@mail.utoronto.ca (J.C.)

**Keywords:** aging, cardiovascular disease, exercise testing, aerobic exercise

## Abstract

While cardiorespiratory fitness (VO_2_peak) can be improved with exercise and training, it is unclear whether older age is associated with an attenuated VO_2_peak improvement among patients with coronary artery disease (CAD) who complete a cardiac rehabilitation (CR) program. A retrospective review of patient demographics and VO_2_peak data from January 2012 to December 2017 was performed. CAD patients were included if they had successfully completed the supervised 6-month CR program (>75% of exercise prescription) and two VO_2_peak assessments (respiratory exchange ratio (RER) >1.0). Among all patients, there was an improvement in VO_2_peak from 21.1 ± 6.3 mL/kg/min to 26.5 ± 7.9 mL/kg/min (+26% ΔVO_2_peak). Patients in the younger age category (age category 1: 30–39 years old) tended to have a greater percent of relative VO_2_peak improvement when compared to all other age categories (e.g., adults 50 years of age and older). In the regression analysis, VO_2_peak improvement was associated with younger age (*β* = −0.286, *p* < 0.0001), after adjustment for the baseline VO_2_peak (*β* = −0.456, *p* < 0.0001), final prescribed exercise speed at CR program completion (*β* = 0.254, *p* < 0.0001), body mass index (*β* = −0.172, *p* < 0.0001), and male sex (*β* = 0.153, *p* < 0.0001). Nonetheless, the study findings indicate that older adults who complete CR may be able to obtain clinically relevant improvements in VO_2_peak of greater than 20%, and therefore, should be referred for CR.

## 1. Introduction

Regular exercise has been recommended for improving cardiorespiratory fitness, muscular strength and endurance, as well as other cardiovascular health outcomes in older adults [1]. Formal cardiac rehabilitation (CR) with exercise programming has been associated with significant improvements in both cardiac risk reduction (e.g., blood pressure, cholesterol level, body composition) and patient outcomes (e.g., higher quality of life, lower hospital readmission rate, and lower mortality rates) [2,3,4]. In fact, cardiorespiratory fitness (measured via peak oxygen consumption (VO_2_peak)) has been shown to improve significantly following a formal CR program [2,5,6]. The importance of a higher VO_2_peak as a “vital sign” [7] cannot be understated, given its strong association with a lower mortality rate [8].

Previous work has shown that younger, male adults are more likely to be referred to and attend a CR program [9,10,11], while older adults who are referred to CR programs tend to be underrepresented and have a lower VO_2_peak at CR program entry [12]. While cardiorespiratory fitness can be improved with exercise and training during the lifespan [13], there are limited data to evaluate the influence of increasing age on improvements in cardiorespiratory fitness following a formal CR program [6,14,15]. In particular, it is unclear whether older age is associated with an attenuated VO_2_peak improvement among patients with coronary artery disease (CAD) and who complete a CR program.

Therefore, the purpose of this study was to determine the association between improvement in cardiorespiratory fitness relative to baseline and increasing age among adults who successfully completed a 6-month formal outpatient CR program. We hypothesize that while there will be a negative association between age and VO_2_peak measured at entry to CR, the relative improvement in VO_2_peak will be significant and similar among all adults with CAD.

## 2. Methods

### 2.1. General Study Procedures

This study was conducted in a large academic CR program at the Toronto Rehabilitation Institute in Toronto, Ontario, Canada. A retrospective review of patient data from January 2012 to December 2017 was performed using the institutional database following institutional research ethics board approval. This institutional database included data on all clinical patients who were referred to the CR program during this time period. Patient demographics and cardiorespiratory fitness assessment data were collected.

### 2.2. Inclusion/Exclusion Criteria

Patients with a history of CAD were included in this analysis if they had successfully completed the supervised 6-month outpatient CR program with a high “compliance” rating (completion of >75% of exercise prescription) and had completed two cardiorespiratory fitness (VO_2_peak) assessments (respiratory exchange ratio (RER) >1.0, indicative of adequate physiologic effort for both tests). Patients were excluded from this analysis if they had received a valve repair and/or a prior diagnosis of heart failure, heart transplant, congestive heart failure, stroke or atrial fibrillation, did not comply with/complete the exercise recommendations (<75% compliant in completing their exercise prescription), and/or were identified as a “non-responder” to cardiac rehabilitation (a decline in VO_2_peak). All data were collected as a part of the usual care process for assessment (cardiopulmonary testing, cardiometabolic risk blood draw) and treatment (exercise prescriptions) that is provided to all patients who enter the program.

### 2.3. Cardiorespiratory Fitness Assessment

VO_2_peak was measured before and after the 6-month CR program using a standardized cardiorespiratory exercise stress-test protocol, as per standard of care. A graded exercise test with gas analysis was performed to determine VO_2_peak (Vmax Series Software Version 12-3A, SensorMedics) using a treadmill (Cardiac Sciences ST55, Waukesha, WI, USA) or cycle ergometer (Ergoselect 200 P; Ergoline GmbbH, Bitz, Germany). The relative improvement in peak cardiorespiratory fitness following the CR program was calculated as a primary outcome variable (net change in VO_2_peak from post-test to pre-test, expressed as a percentage). Body mass index (BMI, kg/m^2^) was also measured.

### 2.4. Cardiac Rehabilitation Exercise Prescription and Program

Aerobic exercise prescriptions were determined based on individual patient VO_2_peak data. All participants completed an aerobic exercise program (progressing to 5 times per week, including 1 supervised, on-site exercise session), equivalent to ~60–75% of their VO_2_peak or heart-rate reserves obtained during their cardiorespiratory fitness assessments. This individualized exercise prescription intensity was consistent with recommendations for patients with CAD [16]. Participants were also instructed to complete a standardized resistance exercise program, using free weights and/or resistance bands (2 times per week), in addition to their aerobic exercise prescription. Participants submitted weekly exercise diaries to document at-home aerobic and resistance exercise sessions.

### 2.5. Statistical Analysis

Pre- and post-CR descriptive statistics were reported for the whole group and by age (per decade). Paired sample *t*-tests (including Bonferroni correction) were used to compare pre- and post-CR variables. An analysis of covariance (ANCOVA) was performed to evaluate between-age category differences in the relative increase in VO_2_peak while controlling for initial VO_2_peak prior to CR. Pearson correlation and stepwise regression analyses were performed to examine the factors associated with a greater percent relative VO_2_peak improvement. Patient age, sex, BMI, initial VO_2_peak, and change in prescribed exercise speed during the CR program were included in the regression model. All analyses were performed using IBM SPSS Statistics 22 (IBM, Armonk, NY, USA). A *p*-value of 0.05 was considered to be significant.

## 3. Results

### 3.1. Patient Demographics

A total of 1450 CAD patients completed a 6-month CR program and met the inclusion criteria for this study. Patient demographics and cardiorespiratory fitness assessment data are presented in Table 1, and by age category (by decade from 30 to 90 years of age) in Table 2. Briefly, trends were observed in patient characteristics with increasing age category, including a decline in BMI, an increase in systolic blood pressure and a decline in cardiorespiratory fitness (VO_2_peak). The exercise prescription in terms of distance, time, and speed was also lower for the older age groups. The CR program completion was associated with a significant improvement in BMI in all patient age categories (except patients in the 30–39 and 40–49-year-old age categories).

### 3.2. Cardiorespiratory Fitness Following CR Program

Among all CAD patients, there was a significant improvement in VO_2_peak from 21.1 ± 6.3 mL/kg/min to 26.5 ± 7.9 mL/kg/min (+26% ΔVO_2_peak; Table 1). VO_2_peak increased in all age groups (Table 2; *p* < 0.05 for all), with the largest absolute changes seen in the younger subjects. Patients in the younger age category (age category 1: 30–39 years old) tended to have a greater percent relative to VO_2_peak improvement when compared to all categories, which included adults older than 50 years of age and up. There was no difference in percent relative to VO_2_peak improvement when comparing patients in the two youngest age categories (age category 1: 30–39 years old versus age category 2: 40–49 years old) and two oldest age categories (age category 5: 70–79 years old vs. age category 6: 80–90 years old). In Pearson *r* correlational analysis, a modest inverse association was observed between relative improvement in VO_2_peak and age (*r* = −0.15, *p* < 0.0001). In regression analysis, a greater percent relative to VO_2_peak improvement was associated with younger age, greater change in prescribed exercise speed (from CR program entry to completion), lower baseline VO_2_peak, lower baseline BMI, and male sex (Table 3). Nonetheless, the proportion of variance that can be explained by the regression model was modest (adjusted *R*^2^ = 0.29).

## 4. Discussion

Significant improvements in cardiorespiratory fitness have been observed previously among patients with coronary artery disease following CR [3,6,12]; however, it is unknown if the magnitude of improvement in cardiorespiratory fitness is attenuated with age, particularly among older adults who are less likely to receive a referral and complete CR [11]. Our retrospective analysis of 1450 patients with CAD revealed two main findings: (1) a greater relative improvement in cardiorespiratory fitness was observed among younger versus older adults who successfully completed a CR program, and yet (2) all age categories, including older age categories (e.g., 70 years and older), experienced a clinically significant increase in relative peak cardiorespiratory fitness of greater than 20%, despite established age-related declines in absolute cardiorespiratory fitness. The strengths of this study included a large sample of patients with CAD who underwent cardiorespiratory fitness with direct gas measurement prior to and following a 6-month cardiac rehabilitation program with individualized exercise prescriptions and healthy lifestyle management.

### 4.1. Cardiorespiratory Fitness, Exercise Training, and Aging

Regular exercise has been associated with improvements in cardiorespiratory fitness across the lifespan [1,3,6,14]. While patients in the younger age category tended to have a significantly greater change in cardiorespiratory fitness, the mean magnitude of improvement was remarkable across the lifespan, ranging from 42% (30–39-year-olds) to 21% (80–90-year-olds). The current study findings also demonstrate significant heterogeneity in the magnitude of improvement in cardiorespiratory fitness both within and between age categories following a well-controlled, supervised exercise intervention that was based on ‘gold-standard’ cardiorespiratory fitness assessment. Our study included strict inclusion and exclusion criteria of adults with stable CAD across the adult lifespan who demonstrated responsiveness to the exercise training program (>0% improvement in VO_2_peak with CR) and supervised by CR staff. These criteria were selected to better understand treatment efficacy and the age-related magnitude of improvement in cardiorespiratory fitness among compliant patients who did not demonstrate a decline in cardiorespiratory fitness in the fourth through ninth decades of life. These heterogeneous findings are similar to previous studies that have reported on chronic adaptations to exercise training among healthy adults [14,15,17]. In a recent study, changes in VO_2_peak, ranging from between −8% to +42%, were observed following 20 weeks of combined aerobic and resistance training in 175 untrained healthy middle-aged adults (40–67 years old) [15]. Similarly, relative improvements in VO_2_peak, ranging between 0% and +58%, were reported following 9–12 months of aerobic exercise training in 110 untrained older adults (60–71 years old), with no age or gender difference [14]. Comparatively, the magnitude of improvement in cardiorespiratory fitness observed following a long-term exercise intervention dodides not seem to differ when comparing our cohort of patients with stable coronary artery disease to these other study cohorts of healthy middle-aged and older adults. These findings may collectively suggest that the trainability of cardiac patients and their capacity for improvement in cardiorespiratory fitness may be quite similar to “healthy” adults of similar age, despite the known presence of cardiac-related disease, possible co-morbid conditions and medication use. The proportion of variance in the regression model was modest (adjusted *R*^2^ = 0.25), suggesting that a number of other factors beyond age, sex, initial cardiorespiratory fitness, and exercise prescription explain the trainability of a patients’ cardiorespiratory fitness (e.g., genetics) [18]. Future studies are warranted to examine the predictors of improvement in cardiorespiratory fitness among cardiac patients, as well as compare the predictors of responsiveness among “responders” vs. “non-responders” who complete CR programs. A previous report has suggested that more individualized exercise doses (volume, intensity, modality) are required to elicit improvements in the cardiorespiratory fitness of select individuals who are initially considered “non-responders” [19].

### 4.2. Clinical Implications in the Cardiac Rehabilitation Setting

The trainability of older adults, even those with cardiac disease, is evidenced by their capacity to perform and respond to both aerobic and resistance exercise [1]. While it is clear that not all CR patients will experience the same magnitude of physiological benefit with an increase in cardiorespiratory fitness, exercise training can reduce cardiometabolic risk and improve functional capacity [1], particularly among community-dwelling older adults who need to perform activities of daily living with independence. Historically, older adults have been less likely to receive referrals and/or complete cardiac rehabilitation [4]. Our findings of CR treatment efficacy serve to highlight the importance of patient referrals to CR across the lifespan. A potential increase of greater than 20% in cardiorespiratory fitness, even later in life during the eighth and ninth decades, may undoubtedly contribute to an independent, healthier lifestyle.

### 4.3. Limitations

This retrospective study included a large sample of CAD patients who were referred and highly compliant in completing a 6-month CR program at a single center in a large urban community, which may not be reflective of all CR patients nor older adults living with cardiac-related disease in the community. While our study included both male and females across their lifespans, older adults and women were underrepresented, likely due to the lower referral, compliance, and/or dropout rates associated with these cohorts [9,11]. Our analysis did not control for genetics [18], medication use, or co-morbid conditions, which may impact patients’ ability to improve cardiorespiratory fitness. While our regression analysis did adjust for baseline VO_2_peak and magnitude of exercise prescription progressions (e.g., change in walking speed during 6-month program), it is evident that younger patients were more likely to have a greater overall training load, which may have contributed to a larger increase in VO_2_peak. While our study investigated treatment efficacy within a structured CR program, we were unable to prospectively evaluate and control for changes in individualized exercise prescriptions; therefore, future randomized controlled trials may address this issue.

## 5. Conclusions

Among CAD patients who demonstrated an improvement in cardiorespiratory fitness, younger adults tended to have greater relative improvements in cardiorespiratory fitness following a formal CR program. While older adults tended to have a lower cardiorespiratory fitness prior to CR, the study findings indicated that older adults who participated in CR still derived clinically relevant improvements in cardiorespiratory fitness of greater than 20%; therefore, older adults with CAD should be referred and included in CR programs that incorporate a formal exercise program.

## Figures and Tables

**Table 1 jcm-08-00310-t001:** Patient demographics and cardiorespiratory fitness assessment results at baseline and following completion of cardiac rehabilitation (*n* = 1450).

Variable of Interest	Pre-CR	Post-CR
Age (years)	62 ± 10	-
Sex (*n*, % males)	1234 (85%)	-
Body Mass Index (BMI, kg/m^2^)	28 ± 4	27 ± 4 *
Prescribed Exercise Distance (kilometers)	2.4 ± 1.6	4.2 ± 1.3 *
Prescribed Exercise Time (hours)	0.43 ± 0.17	0.68 ± 0.27 *
Resting HR (beats per minute)	68 ± 12	66 ± 11 *
Resting SBP (mm Hg)	124 ± 16	123 ± 15 *
Resting DBP (mm Hg)	75 ± 9	74 ± 9
Peak HR (beats per minute)	124 ± 31	134 ± 22 *
Peak SBP (mm Hg)	168 ± 24	174 ± 23 *
Peak DBP (mm Hg)	76 ± 10	75 ± 9 *
Peak Oxygen Consumption (mL/kg/min)	21.1 ± 6.3	26.5 ± 7.9 *
Peak Respiratory Exchange Ratio	1.15 ± 0.08	1.17 ± 0.08 *
Total Cholesterol (mmol/L)	3.35 ± 0.84 (*n* = 928)	3.40 ± 0.81 (*n* = 572)
High-Density Lipoprotein (mmol/L)	1.11 ± 0.30	1.28 ± 1.24 *
Low-Density Lipoprotein (LDL) Cholesterol (mmol/L)	1.68 ± 0.64 (*n* = 917)	1.66 ± 0.63 (*n* = 562)
Triglyceride (mmol/L)	1.29 ± 0.64 (*n* = 686)	1.17 ± 0.62 (*n* = 357)
Fasting Blood Glucose (mmol/L)	5.76 ± 1.18 (*n* = 833)	5.78 ± 1.28 (*n* = 547)
Glycosylated Hemoglobin (A1C, %)	5.9 ± 0.7 (*n* = 716)	5.9 ± 0.7 (*n* = 466)

Abbreviations: CR = cardiac rehabilitation, HR = heart rate, SBP = systolic blood pressure, DBP = diastolic blood pressure; * *p* < 0.05.

**Table 2 jcm-08-00310-t002:** Demographics and cardiorespiratory fitness at baseline and following cardiac rehabilitation completion.

Age Category	1 (30–39 Years Old) (*n* = 30)	2 (40–49 Years Old) (*n* = 129)	3 (50–59 Years Old) (*n* = 392)	4 (60–69 Years Old) (*n* = 541)	5 (70–79 Years Old) (*n* = 292)	6 (80–90 Years Old) (*n* = 66)
Patient Demographics						
Age (years)	35.9 ± 2.9	45.5 ± 3.1	55.1 ± 2.8	64.6 ± 2.8	73.8 ± 2.7	82.6 ± 2.5
Sex, *n* (% males)	27 (90)	118 (92)	349 (89)	460 (85)	228 (78)	54 (82)
Cardiorespiratory Fitness Assessment Prior to Cardiac Rehabilitation (Pre-CR)						
Body Mass Index (kg/m^2^)	28.7 ± 9.2	28.1 ± 5.0	27.7 ± 4.4	27.6 ± 4.3	27.0 ± 4.2	26.1 ± 3.5
Resting HR (beats per minute)	70 ± 12	71 ± 12	69 ± 12	68 ± 12	67 ± 12	65 ± 12
Resting SBP (mm Hg)	113 ± 13	116 ± 15	121 ± 15	125 ± 15	128 ± 17	131 ± 17
Resting DBP (mm Hg)	74 ± 10	76 ± 9	76 ± 9	75 ± 9	73 ± 9	71 ± 9
Prescribed Exercise Distance (kilometers)	2.85 ± 1.28	2.60 ± 0.96	2.66 ± 1.88	2.38 ± 1.79	1.99 ± 0.94	1.68 ± 0.70
Prescribed Exercise Time (hours)	0.66 ± 0.15	0.76 ± 0.64	0.89 ± 0.18	0.69 ± 0.20	0.64 ± 0.22	0.58 ± 0.19
Prescribed Exercise Speed (km/hour)	5.78 ± 0.72	5.69 ± 0.79	5.50 ± 0.70	5.23 ± 0.78	4.73 ± 0.93	4.22 ± 0.83
Peak HR (beats per minute)	141 ± 22	136 ± 19	131 ± 20	122 ± 19	113 ± 19	101 ± 17
Peak SBP (mm Hg)	160 ± 23	163 ± 27	170 ± 21	169 ± 23	168 ± 26	164 ± 22
Peak DBP (mm Hg)	75 ± 8	78 ± 10	78 ± 10	79 ± 10	78 ± 9	71 ± 10
Peak Oxygen Consumption (mL/kg/min)	26.3 ± 8.9	24.6 ± 6.0	23.3 ± 6.2	21.0 ± 5.7	17.6 ± 5.3	15.4 ± 3.9
Peak Respiratory Exchange Ratio	1.16 ± 0.08	1.16 ± 0.08	1.15 ± 0.07	1.16 ± 0.10	1.14 ± 0.09	1.13 ± 0.07
Cardiorespiratory Fitness Assessment Following Cardiac Rehabilitation (Post-CR)						
Body Mass Index (kg/m^2^)	28.1 ± 8.2	27.8 ± 5.0	27.5 ± 4.6	27.4 ± 4.3	26.7 ± 3.7	25.7 ± 3.1
Resting HR (beats per minute)	70 ± 11	69 ± 11	67 ± 12	65 ± 11	65 ± 11	62 ± 10
Resting SBP (mmHg)	118 ± 11	116 ± 14	121 ± 15	124 ± 15	127 ± 15	127 ± 17
Resting DBP (mmHg)	75 ± 8	75 ± 9	76 ± 8	74 ± 9	72 ± 9	70 ± 9
Prescribed Exercise Distance (kilometers)	4.53 ± 1.22	4.60 ± 1.03	4.38 ± 1.09	4.15 ± 1.16	3.53 ± 1.26	2.90 ± 0.95
Prescribed Exercise Time (hours)	0.47 ± 0.21	0.45 ± 0.14	0.46 ± 0.17	0.43 ± 0.16	0.40 ± 0.16	0.38 ± 0.13
Prescribed Exercise Speed (km/hour)	6.56 ± 1.05	6.43 ± 0.91	6.12 ± 0.92	5.71 ± 0.93	5.08 ± 0.94	4.55 ± 0.682
Peak HR (beats per minute)	158 ± 15	152 ± 17	143 ± 18	132 ± 19	125 ± 55	106 ± 16
Peak SBP (mm Hg)	160 ± 23	171 ± 24	177 ± 23	175 ± 22	173 ± 22	166 ± 25
Peak DBP (mm Hg)	74 ± 9	76 ± 9	77 ± 8	78 ± 10	77 ± 9	85 ± 11
Peak Respiratory Exchange Ratio	1.19 ± 0.09	1.17 ± 0.08	1.17 ± 0.08	1.17 ± 0.08	1.16 ± 0.08	1.17 ± 0.08
Peak Oxygen Consumption (mL/kg/min)	35.9 ± 8.9 *	32.3 ± 7.6 *	29.5 ± 7.4 *	26.3 ± 6.8 *	21.3 ± 5.8 *	18.4 ± 4.2 *
Percent Change in Relative VO_2peak_	42 ± 31	35 ± 31	29 ± 24	27 ± 23	24 ± 21	21 ± 18

Abbreviations: CR = cardiac rehabilitation, HR = heart rate, SBP = systolic blood pressure, DBP = diastolic blood pressure. * *p* < 0.05 for change in VO_2_peak from pre-CR to post-CR.

**Table 3 jcm-08-00310-t003:** Factors associated with greater relative improvements in cardiorespiratory fitness among all Coronary Artery Disease (CAD) patients.

Characteristics (*n* = 1450)	Mean ± SD or %	Association with ΔVO_2_peak (*β*)	Sig. (*p*)
Baseline VO_2_peak (mL/kg/min)	21.1 ± 6.3	−0.456	0.000
Change in Prescribed Exercise Speed (km/hr)	0.1 ± 0.1	0.254	0.000
Age (years)	62 ± 10	−0.286	0.000
BMI (kg/m^2^)	28 ± 4	−0.172	0.000
Male Sex (%)	85%	0.153	0.000

Abbreviations: *β* = Standardized coefficient beta, SD = standard deviation, BMI = body mass index.

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
