# Peer review of "Age-Related Improvements in Peak Cardiorespiratory Fitness among Coronary Heart Disease Patients Following Cardiac Rehabilitation"

_jcm, 2019, doi:10.3390/jcm8030310_

Reviewer 1 Report

This article, entitled “Age-related improvements in peak cardiorespiratory fitness among coronary heart disease patients following cardiac rehabilitation”, retrospectively reviewed theVO2 peak improvement after an aerobic exercise program among patients with coronary heart disease. Authors concluded that younger adults tended to have greater improvements after a cardiac rehabilitation program in cardiorespiratory fitness than older one. This topic is interesting, but some queries should be clarified.

Major queries

In this retrospective study, authors should demonstrate the differences in cardiorespiratory fitness training across the age categories. It is reasonable to think a greater intensity of exercise training in young patients than that in old one.

Since the cardiac rehabilitation had persisted for 6 months, the data of BMI after the program should be showed. If BMI was not assessed after cardiac rehabilitation program, it should be discussed.

The correlation coefficient of ‒0.15 is relative small for clinical association. A completed table is necessary for regression analysis. Author should explain what the β means, and why baseline BMI was not included. 

Minor queries

The number of male patients was not shown in Table 2.

The units of lipids and glucose should be mmol/L.

Author Response

See attached file for response to reviewer comments.

Reviewer 2 Report

In this manuscript, the purpose of this study was to determine the association between improvement in cardiorespiratory fitness relative to baseline and increasing age among adults who successfully completed a 6-month formal outpatient CR program.

In order to do this, a retrospective review of patient demographic and VO2peak data from January 2012 – December 2017 was performed.

The results revealed that among CAD patients who demonstrated an improvement in cardiorespiratory fitness, younger adults tended to have greater relative improvements in cardiorespiratory fitness following a formal CR program, while the older adults who participate in CR still derive clinically-relevant improvements in cardiorespiratory fitness of greater than 20%.

So the older adults with CAD should be referred and included in CR programs that incorporate a formal exercise program.

For the study the presented data are quite sufficient.

Author Response

See attached file for response to reviewer comments.

Round  2

Reviewer 1 Report

Authors should explain or discuss the effects of greater exercise-training intensity in young patients than that in old ones. This difference is an important bias for the conclusion.

The data are not consistent between Table 1 and Table 3.

Author Response

Thank you for your comments. We have corrected Table 3 with the appropriate mean values. We have added a statement in the limitations section to comment on the issue of exercise training volume in the younger vs. older CR patients. It reads: "While our regression analysis did adjust for baseline VO2peakand magnitude of exercise prescription progressions (e.g. change in walking speed during 6 month program), it is evident that younger patients were more likely to have a greater overall training load, which may have contributed to a larger increase in VO2peak.While our study investigated treatment efficacy within a structured CR program, we were unable to prospectively evaluate and control for changes in individualized exercise prescriptions; therefore, future randomized controlled trials may address this issue."